# Risk Factors for Infectious Complications following Endoscopic Retrograde Cholangiopancreatography in Liver Transplant Patients: A Single-Center Study

**DOI:** 10.3390/jcm13051438

**Published:** 2024-03-01

**Authors:** Norman Kühl, Richard Vollenberg, Jörn Arne Meier, Hansjörg Ullerich, Martin Sebastian Schulz, Florian Rennebaum, Wim Laleman, Neele Judith Froböse, Michael Praktiknjo, Kai Peiffer, Julia Fischer, Jonel Trebicka, Wenyi Gu, Phil-Robin Tepasse

**Affiliations:** 1University of Münster, 48149 Münster, Germany; norman.kuehl@ukmuenster.de; 2Department of Medicine B for Gastroenterology, Hepatology, Endocrinology and Clinical Infectiology, University Hospital Münster, 48149 Münster, Germany; richard.vollenberg@ukmuenster.de (R.V.); joernarne.meier@ukmuenster.de (J.A.M.); hansjoerg.ullerich@ukmuenster.de (H.U.); martinsebastian.schulz@ukmuenster.de (M.S.S.); florian.rennebaum@ukmuenster.de (F.R.); wim.laleman@uzleuven.be (W.L.); michael.praktiknjo@ukmuenster.de (M.P.); kai-henrik.peiffer@ukmuenster.de (K.P.); julia.fischer2@ukmuenster.de (J.F.); jonel.trebicka@ukmuenster.de (J.T.); wenyi.gu@ukmuenster.de (W.G.); 3Department of Gastroenterology and Hepatology, University Hospitals Leuven, KU Leuven, 3000 Leuven, Belgium; 4Institute of Medical Microbiology, University Hospital Muenster, 48149 Münster, Germany; neelejudith.froboese@ukmuenster.de

**Keywords:** liver transplantation, ERCP, cholangitis, risk factors, infection, ischemic type biliary lesions, ITBL, antibiotic therapy

## Abstract

**Background**: Liver transplant recipients often require endoscopic retrograde cholangiopancreatography (ERCP) for biliary complications, which can lead to infections. This retrospective single-center study aimed to identify risk factors for infectious complications following ERCP in liver transplant patients. **Methods**: A retrospective analysis was conducted on 285 elective ERCP interventions performed in 88 liver transplant patients at a tertiary care center. The primary endpoint was the occurrence of an infection following ERCP. Univariable and multivariable regression analyses, Cox regression, and log-rank tests were employed to assess the influence of various factors on the incidence of infectious complications. **Results**: Among the 285 ERCP interventions, isolated anastomotic stenosis was found in 175 cases, ischemic type biliary lesion (ITBL) in 103 cases, and choledocholithiasis in seven cases. Bile duct interventions were performed in 96.9% of all ERCPs. Infections after ERCP occurred in 46 cases (16.1%). Independent risk factors for infection included male sex (OR 24.19), prednisolone therapy (OR 4.5), ITBL (OR 4.51), sphincterotomy (OR 2.44), cholangioscopy (OR 3.22), dilatation therapy of the bile ducts (OR 9.48), and delayed prophylactic antibiotic therapy (>1 h after ERCP) (OR 2.93). Additionally, infections following previous ERCP interventions were associated with an increased incidence of infections following future ERCP interventions (*p* < 0.0001). **Conclusion**: In liver transplant patients undergoing ERCP, male sex, prednisolone therapy, and complex bile duct interventions independently raised infection risks. Delayed antibiotic treatment further increased this risk. Patients with ITBL were notably susceptible due to incomplete drainage. Additionally, a history of post-ERCP infections signaled higher future risks, necessitating close monitoring and timely antibiotic prophylaxis.

## 1. Introduction

Numerous complications following liver transplantation exert an impact on the frequency of patient hospitalizations and mortality rates [1]. Incomplete biliary drainage in liver transplant patients poses significant risks, including heightened susceptibility to infections due to stagnant bile. It also increases the likelihood of cholangitis, marked by symptoms like fever and in some cases severe infection and sepsis. Scar formation in the bile ducts can impede normal bile flow, contributing to long-term liver function impairment. Ultimately, these factors may culminate in liver damage, elevating the risk of complications such as cirrhosis and failure.

After liver transplantation, biliary complications, such as strictures in the area of the bile duct anastomosis, as well as ischemic type biliary lesions (ITBL), are common, and often require intervention; early detection and effective management of incomplete biliary drainage are vital for minimizing risks and improving overall patient outcomes. Endoscopic retrograde cholangiopancreatography (ERCP) is considered the primary approach for managing these complications. However, ERCP procedures themselves can lead to complications even in non-transplant patients, including post-ERCP pancreatitis, perforation of the bowel, and bleeding. Infectious complications are rare in non-transplant patients, but transplant patients show heightened susceptibility to infections, coupled with the potential for cholangitis and septic complications, underscoring the critical importance of early detection and effective management of infections [2,3,4]. The definitions of infectious complications vary, with some criteria including fever, increased C-reactive protein levels, and positive blood cultures. Previous studies have reported infectious complication rates of 0.5–3% in non-transplant patients, with risk factors being sphincterotomy, stenting of malignant stenoses, hilar obstruction, inadequate biliary drainage, and cholangioscopy-associated irrigation [5,6,7,8,9]. Patients with incomplete biliary drainage after ERCP, such as those with primary sclerosing cholangitis or cholangiocarcinoma, and liver transplant patients are at higher risk for infectious complications [3,10,11]. Prophylactic antibiotic therapy is commonly used to prevent infections in various medical fields (for example in liver transplantation [12]). Its efficacy in ERCP procedures remains inconsistent. Some studies suggest a protective effect against transient bacteremia, but its impact on the development of cholangitis and sepsis is unclear [13,14,15,16,17,18,19,20]. There are concerns about the potential for enhancing multidrug-resistant organisms (MDROs) that may lead to duodenoscope-related infections and outbreaks [21]. Currently, the American and European Societies for Gastrointestinal Endoscopy (ASGE and ESGE) recommend prophylactic antibiotic therapy for non-transplant patients with unlikely complete bile duct drainage and all liver transplant patients undergoing ERCP. Furthermore, for defined interventions such as the placement of a gastrostomy, transgastric cyst puncture, or cholangioscopy, the professional societies recommend the administration of prophylactic peri-procedural antibiotic therapy, even in nontransplant patients [15,22]. However, it is uncertain whether all liver transplant patients require antibiotic therapy or if certain factors within this patient cohort predispose them to infections. This retrospective study aims to identify risk factors for the development of infectious complications after ERCP in a cohort of liver transplant patients. The findings could potentially help tailor prophylactic measures and improve patient outcomes in this vulnerable population.

## 2. Materials and Methods

This study was conducted as a retrospective single-center investigation at Muenster University Hospital, Germany, a tertiary care center. The research aimed to analyze cases of endoscopic retrograde cholangiopancreatography procedures in liver transplant patients within the timeframe of January 2017 to February 2021, as identified through the digital hospital information system. Patients who underwent elective ERCP procedures after liver transplantation and received periprocedural antibiotic therapy before, during, or after the procedure were considered for inclusion. The indications for elective ERCP procedures are shown in the results section. Emergency patients displaying signs of acute infection before ERCP (elevated C-reactive protein (CRP) levels upper the normal value and fever so that on this basis an acute infection had to be assumed before ERCP) or experiencing complications other than infection after ERCP (e.g., pancreatitis, bleeding, or perforation) were excluded. A thorough analysis of electronic medical records was conducted to gather predefined data, including patient age, sex, date of transplantation, date of ERCP, time interval between transplantation and ERCP (measured in months), indication for transplantation, ERCP findings, procedures performed during ERCP, details of antibiotic therapy (preparation and timing), blood culture results (taken only in cases of post-interventional fever), C-reactive protein CRP-values, and leukocyte counts after ERCP. The primary outcome measure for this study was the occurrence of an infection after ERCP. Infection was defined as the development of fever exceeding 38.5 °C and an increase in CRP levels after ERCP, leading to the initiation of antibiotic therapy within three days following the procedure. This definition was based on previously published criteria [9,10,11]. The ERCP procedures were conducted in an operating room compliant with local hygiene regulations. If necessary, sterile disposable materials were utilized. The processing of the endoscopes was also carried out in accordance with local legal requirements. The study received ethical approval from the local ethics committee at Muenster University Hospital (Approval number: 2021-460-f-S).

### Statistical Analyses

Categorical variables were presented as absolute numbers and percentages, and their comparisons were performed using either the chi-square test or Fisher’s exact test, depending on the sample size and data distribution (the chi-square test was applied when the underlying distribution of the data deviated from normality). Continuous variables were expressed as medians with interquartile ranges (IQR) or minimum/maximum values, and their comparisons were conducted using either Student’s *t*-test or the Mann–Whitney U test, based on the distribution of the variable (the Mann–Whitney U test was chosen when the distribution of the variable deviated from normality). Initially, procedure-based risk factors for the development of infectious complications after each ERCP (dependent variable) were investigated. Univariable logistic regression analysis was applied to examine the potential risk factors independently. Subsequently, multivariable logistic regression modeling was employed to identify independent factors influencing the development of infectious complications. Non-metric variables were treated as categorical variables in the model. Next, patient-based analyses were carried out. The risk of developing post-ERCP infections within a two-year follow-up period was compared between patients with or without infections after the first ERCP recorded during the study period, utilizing Kaplan–Meier curves and log-rank tests. To account for different numbers of ERCPs within the two-year follow-up, a multivariable Cox regression model was conducted to identify independent risk factors for post-ERCP infection development in the cohort of 88 patients. Patients who had only one recorded ERCP and had follow-up periods of less than two years were considered as lost to follow-up and were censored in the analysis. All statistical tests were two-sided, and a *p*-value less than 0.05 was considered statistically significant. The statistical analyses were performed using SPSS version 26 (IBM, Chicago, IL, USA).

## 3. Results

### 3.1. Cohort Characteristics

Initially, 403 ERCP procedures in liver transplant patients were identified. Out of these, 118 procedures were excluded from the study due to missing information in patient records (*n* = 84), signs of infection (elevation of CRP levels and fever) before ERCP (*n* = 22), or occurrence of a complication other than infection after ERCP (eight cases due to clinical signs of pancreatitis, two cases due to bleeding, two cases due to suspected perforation of the bile ducts after dilatation therapy). Ultimately, a total of 285 ERCP procedures (involving 88 patients) were included in the study (see Figure 1). The median age of the patients at the time of ERCP was 56 years (ranging from 21 to 77 years), with 27 patients (30.7%) being female. ERCP procedures were performed at a median of 17 months (ranging from 0 to 226 months) after liver transplantation. The median duration of hospitalization was 4 days (ranging from 2 to 23 days). The respective indications for liver transplantation and other cohort characteristics are provided in Table 1.

### 3.2. ERCP Procedures

A total of 285 elective ERCP procedures were performed, with each patient undergoing a median of two ERCP procedures. The indication for ERCP was related to progressively elevated liver tests (gamma-glutamyltransferase, bilirubin, alkaline phosphatase) in 243 cases and/or suggestive findings of biliary obstruction on non-invasive imaging in 118 cases (MRI, CT, ultrasound). In seven cases bile stones were diagnosed before ERCP by ultrasound. Among the ERCP procedures, 175 (61.4%) involved a singular biliary-anastomosis, 103 (36.1%) revealed ischemic type biliary lesions (ITBL), and seven (2.5%) identified (sub)obstructive choledocholithiasis without signs of strictures or ITBL. Therapeutic interventions were conducted in 276 ERCP procedures (96.8%), while nine procedures (3.2%) did not require any intervention due to only discrete narrowing of the biliary anastomosis (Table 2).

### 3.3. Infections and Anti-Infective Therapy

Infections following the ERCP procedure were found in 46 out of 285 cases (16.1%), involving 25 out of 88 patients. Blood cultures were obtained in 31 out of 46 infection cases (67.4%) within 24 h after the ERCP procedure due to clinical signs of infection (fever), with 10 of these cases (32.3%) showing a positive blood culture. The most frequently isolated bacteria were *E. faecium* (30%), *E. coli* (20%), and (20%). No multidrug-resistant pathogens were identified in the blood cultures, and no cases of septic course or liver abscess were reported (Table 3). Peri-interventional prophylactic antibiotic therapy was administered in 188 cases (66%) up to one hour before the start of ERCP or during the procedure, while in 97 cases (34%), it was delayed (defined as administered more than one hour after ERCP). Piperacillin/tazobactam was the most commonly administered prophylactic antibiotic in 210 cases (73.7%), with ceftriaxone (12%), meropenem (5.3%), ciprofloxacin (3.5%), and other substances (5.5%) being used in cases of penicillin allergy or corresponding combined intolerances (Table 3). Notably, 23 out of the 46 infection cases (50%) received delayed antibiotic prophylactic therapy compared to 74 out of 240 cases (31%) without post-procedure infection (Fisher’s exact test; *p* = 0.017). Additionally, 42 patients (47.7%) were colonized by multidrug-resistant bacteria at the time of the first ERCP procedure during the study period.

### 3.4. Identification of Risk Factors for Infectious Complications after ERCP

Univariable logistic regression analysis was utilized to investigate the influence of individual factors on the development of infectious complications after ERCP. The following factors were associated with an increased risk of infections: male sex (OR 9.62, *p* = 0.002), prednisolone therapy (OR 3.18, *p* = 0.01), ITBL (OR 3.71, *p* < 0.001), cholangioscopy (OR 2.89, *p* = 0.007), and delayed application of prophylactic anti-infective therapy (more than 1 h after ERCP) (OR 2.23, *p* = 0.01). Patients with infectious complications after ERCP had significantly higher median CRP levels (1 mg/dL vs. 6.55 mg/dL, *p* < 0.001) and median leukocyte counts (4.95 Tsd/μL vs. 6.27 Tsd/μL, *p* < 0.001) after ERCP compared to patients without infections, and they were hospitalized significantly longer (4 days vs. 5 days, *p* = 0.003) (Table 4). Multivariable logistic regression analysis was then conducted to assess independent risk factors for the development of infectious complications after ERCP. The analysis revealed that the following factors independently influenced the risk of infection: male sex (OR 24.19, *p* < 0.001), prednisolone therapy (OR 4.5, *p* = 0.006), ITBL (OR 4.51, *p* < 0.001), sphincterotomy (OR 2.44, *p* = 0.04), cholangioscopy (OR 3.22, *p* = 0.02), and delayed application of prophylactic anti-infective therapy (more than 1 h after ERCP) (OR 2.93, *p* = 0.006) (Table 5). The use of the antibiotic preparation had no influence on the development of an infection.

### 3.5. Risk Factors for Subsequent Infections following ERCP

To further investigate the potential independent risk factors for the development of infectious complications after ERCP in different patients, the cumulative incidence of post-ERCP infection was compared between patients who had an infection after the first documented ERCP and those without an infection during a two-year follow-up. Remarkably, patients with infections after the first ERCP documented in the study period showed a significantly higher risk of further development of post-ERCP infections (89.1% vs. 48.1%, log-rank *p* < 0.0001) (Figure 2). In line with the univariable and multivariable analyses in the ERCP cases, a Cox regression model was conducted in 88 patients within a two-year follow-up to identify other potential confounding factors in the development of infectious complications after ERCP. This analysis demonstrated that the development of infections after the first documented ERCP during the study period was the only independent risk factor for the subsequent development of infections within two years of the first ERCP (Table 6).

## 4. Discussion

The present study is the first to explore individual risk factors for the development of infections after ERCP in a cohort of liver transplant patients. Within our study, 16.1% of cases exhibited infections after ERCP, defined by the presence of fever and an increase in CRP levels leading to antibiotic therapy. Notably, this infection rate appears higher compared to other studies [5,19,23], which have reported infection rates ranging from 0.2% to 5% in liver transplant patients after ERCP. It is essential to acknowledge that the definition of infection after ERCP in the existing literature is highly inconsistent, limiting comparability in this domain. The relatively elevated infection rate in our study might be attributed, in part, to the substantial proportion of interventions involving ITBL (36%). Multivariate regression analysis in our study highlighted the presence of ITBL as a highly significant independent risk factor for infection development compared to the presence of only a singular anastomotic stenosis (OR 4.51, *p* < 0.001). This observation could be attributed to the presence of multiple stenoses within the bile duct system as a consequence of ITBL and the accompanying cast formation, which may impede proper and complete biliary drainage after ERCP, in contrast to singular anastomotic stenosis, which usually no longer represents a relevant obstruction to bile drainage following dilatation and stenting. The inability to restore adequate biliary drainage after the injection of contrast media into obstructed bile ducts during ERCP represents one of the key risk factors for post-ERCP infection [18,24]. A study by Wobser et al. examining infection development after ERCP in high-risk patients with primary and secondary sclerosing cholangitis (PSC, SSC) [25] reported that 14.3% of patients not receiving antimicrobial agents developed an infection, which aligns with our findings. In PSC, SSC, and ITBL patients, multiple stenoses in the biliary system frequently lead to incomplete biliary drainage after ERCP.

The incidence of bacteremia following ERCP procedures can be as high as 15% in diagnostic ERCP procedures and up to 26% in therapeutic ERCP procedures. However, the occurrence of infectious adverse events post-ERCP, such as sepsis or cholangitis, is reported to be between 0.5% and 10%. For cholangioscopy, a bacteremia rate of 8.8% and a cholangitis rate of 7% has been found [26,27,28]. In our study, cholangioscopy emerged as a significant risk factor in our cohort (OR 3.22, *p* = 0.02), in agreement with findings in non-transplanted patients from a prior study [9]. Consequently, the ESGE recommends prophylactic antibiotic therapy for every cholangioscopy, regardless of the presence of immunosuppressive therapy [22]. Sphincterotomy is widely recognized as a common risk factor for complications such as bleeding and pancreatitis after ERCP [29]. In our study, sphincterotomy exhibited a significant influence on the development of infectious complications (OR 2.44, *p* = 0.04), corroborating previous findings by Chen et al., who identified endoscopic sphincterotomy as an independent risk factor for post-ERCP cholangitis in an unselected patient cohort [8]. Additionally, our study identified dilatation therapy of the bile ducts as another risk factor (OR 9.48, *p* = 0.03), and to our knowledge, comparable study results in this context are currently unavailable. Taken together, manipulation of the bile ducts, particularly cholangioscopy, dilatation therapy, and sphincterotomy, should be considered in risk assessment and the decision to employ antibiotic prophylactic therapy in liver transplant patients.

In our study cohort, prednisolone emerged as an independent risk factor for infection development among the immunosuppressants used. As far as we know, no study has explicitly investigated the influence of prednisolone on the infection rate in ERCP. Notably, in our study cohort, prednisolone was consistently used as a third immunosuppressant, alongside combinations of tacrolimus, everolimus, and mycophenolate mofetil. Patients receiving prednisolone therapy underwent triple immunosuppression, rendering them generally more immunosuppressed than those receiving one or two immunosuppressives, potentially rendering them more susceptible to bacterial infections after ERCP. However, it is known that patients on glucocorticoids are more prone to a range of infections, such as pneumonia and urinary tract infections [30], as well as opportunistic infections [31,32]. Therefore, it is conceivable that glucocorticoids also increase the risk of bacterial infection after ERCP. Particularly in triple immunosuppressed patients, antibiotic prophylactic therapy should be considered, with stringent monitoring.

In our study cohort, a significant proportion of patients (47.7%) exhibited colonization with multidrug-resistant bacteria at the time of the first ERCP. The significance of colonization with multidrug-resistant pathogens is often not unequivocally elucidated in many instances. For example, the state of research among liver transplant recipients remains inconclusive, and the impact on post-transplant survival varies across different studies [33,34]. However, our analyses did not demonstrate any influence of multidrug-resistant bacterial colonization on the infection rate after ERCP procedures.

Additionally, our study was the first to investigate whether the occurrence of infections after ERCP affects the incidence of further infections after subsequent ERCP interventions. We found that patients who developed an infection after the initial ERCP were significantly more likely to experience infections following subsequent ERCP interventions compared to patients who had not previously developed infections. Our study thus provided the first evidence that past infections indicate an increased risk of re-infection after future ERCP interventions. Consequently, patients with a history of infection should be monitored particularly vigilantly after interventions, and prospective studies should explore whether prolonged antibiotic therapy can prevent infections in these cases. In addition, the influence of duodenoscope-associated infections should be addressed. It has been recognized for several years that despite thorough disinfection of reusable duodenoscopes, a significant proportion of these instruments remain colonized with pathogens that can be transmitted to the next patient during subsequent endoscopic procedures [35]. This circumstance can pose a substantial risk, particularly for liver transplant recipients, wherein the susceptible biliary pathways can become colonized with pathogens during ERCP. As a potential solution, the implementation of single-use duodenoscopes has been deliberated upon, obviating the potential for pathogen transmission [21]. While our study does not contribute further insights to this matter, it should serve as an impetus for the exploration of this subject through prospective investigations.

As liver transplant patients represent a high-risk population for infection after ERCP interventions, the ASGE and ESGE guidelines recommend prophylactic administration of antibiotic therapy regardless of individual risk factors, despite insufficient data [15,22]. However, whether there are specific risk factors or vulnerable subgroups in this cohort remains unexplored, and prospective studies are lacking. Given that antibiotic therapy usage contributes to the development of resistance in many bacterial species, its use should be limited as much as possible. A retrospective study demonstrated a significant increase in resistant bacteria in the bile of patients who received antibiotic therapy during ERCP [36]. Another study found bacterial resistance in blood cultures to antibiotics previously used as prophylactic therapy during ERCP [37]. In our study, we also assessed the impact of systemically effective peri-interventional antibiotic prophylactic therapy. Overall, the therapy did not influence the incidence of infections after ERCP (OR 5.00, CI 0.66–37.92; *p* = 0.12), confirming results from further studies on unselected patient cohorts [26,27,28] and a retrospective study in liver transplant patients, which, however, excluded patients with incomplete biliary drainage after ERCP [13] and lacked a prospective approach. We also investigated the influence of delayed antibiotic administration, 1 h or later after ERCP. These patients did not receive prophylactic antibiotic therapy before or during the intervention but received it later than 1 h after ERCP. Importantly, there were no signs of infection at the time of antibiotic administration, and the therapy was conducted with a prophylactic intent. Delayed administration of antibiotic prophylactic therapy during ERCP examinations occurred more frequently in our endoscopy department before the standard operating procedure for antibiotic therapy administration was optimized within the study period. Both univariable regression analysis (OR 2.23, *p* = 0.01) and multivariable regression analysis (OR 2.93, *p* = 0.006) identified delayed administration as a significant risk factor for infections in our study cohort. Consequently, delayed antibiotic therapy application in liver transplant patients clearly emerged as an independent risk factor for infections, underscoring the necessity of antibiotic prophylactic therapy in these patients and highlighting the importance of early, preferably pre- or intra-interventional application.

It is essential to acknowledge some limitations in our study. The criteria for defining an infection after ERCP are not uniform. This complicates the classification of the data from our study in the existing literature. Therefore, we recommend the standardization of definitions for future research. The study provides limited information on the influence of antibiotic therapy in general, as the proportion of patients without antibiotic therapy was too small to make comprehensive statistical statements. Furthermore, due to the small number of patients in certain subgroups, particularly ITBL patients, we were unable to investigate the influence of antibiotic therapy in these specific subgroups. Therefore, larger, preferably prospective studies are necessary. Moreover, in the Cox regression model, the exclusion of patients with only one ERCP or less than a two-year follow-up period may lead to a bias in the results, as potentially healthier patients who did not require a repeat ERCP may have been disproportionately excluded and additionally, some ERCP procedures conducted shortly after the end of the observation period may not have been taken into account. Lastly, the retrospective design of the study may introduce inaccuracies in individual parameter assessment, particularly in the detailed assessment of infectious complication occurrence.

Nevertheless, our study contributes important insights and aids in the identification of individuals at higher risk within the already vulnerable liver transplant patient cohort. The efficacy of prophylactic antibiotic therapy in the individual patient groups, especially ITBL patients, requires further exploration through prospective studies.

## 5. Conclusions

In liver transplant patients undergoing ERCP, male sex, prednisolone therapy, and complex bile duct interventions independently raised infection risks. Delayed antibiotic treatment further increased this risk. Patients with ITBL were notably susceptible due to incomplete drainage. Additionally, a history of post-ERCP infections signaled higher future risks, necessitating close monitoring and timely antibiotic prophylaxis.

## Figures and Tables

**Figure 1 jcm-13-01438-f001:**
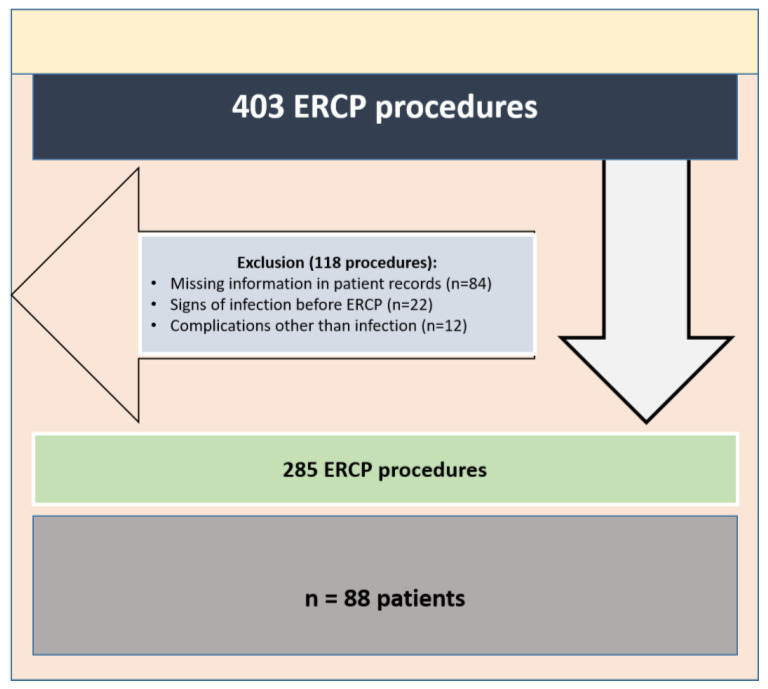
Study flow chart.

**Figure 2 jcm-13-01438-f002:**
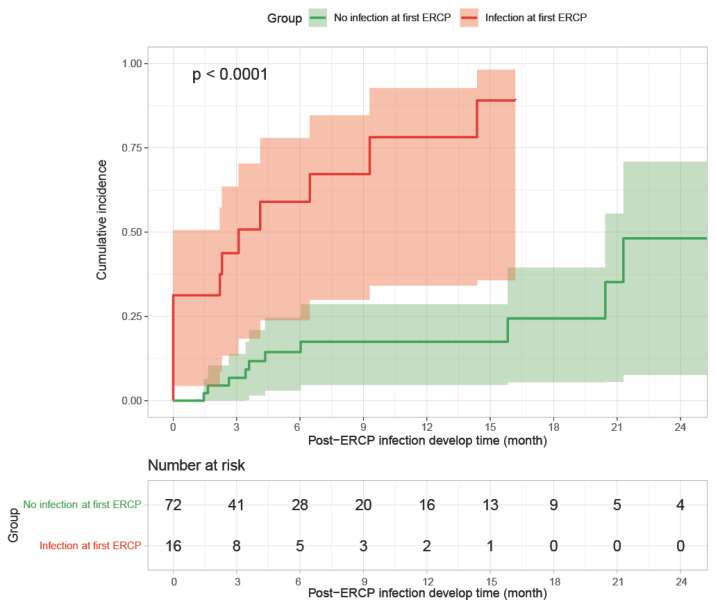
Comparison of the risk of developing post-ERCP infections within two years between patients with or without infections after the first ERCP; Kaplan–Meier curve and log-rank test.

**Table 1 jcm-13-01438-t001:** Patient characteristics and procedure details.

Characteristics (*n* = 88 Patients, *n* = 285 ERCP Procedures)	
Sex, female, absolute (%)	27 (30.7)
Age at time of ERCP, years, median (min./max.)	56 (21/77)
Period from transplant to ERCP, month, median (min./max.)	17 (0/226)
Length of hospital stay, days, median (min./max.)	4 (2/23)
Biliodigestive anastomosis, absolute (%)	6 (6.8)
MDR colonization, absolute (%)	42 (47.7)
Indication for liver transplantation, absolute (%)	
Alcoholic liver cirrhosis	16 (18.2)
Primary sclerosing cholangitis	15 (17.0)
Non-alcoholic steatohepatitis	12 (13.6)
Hepatocellular carcinoma	12 (13.6)
Cryptogen	9 (10.2)
Medicinal/toxic	5 (5.7)
Autoimmune hepatitis	5 (5.7)
Viral hepatitis	4 (4.5)
Wilson’s disease	4 (4.5)
Primary biliary cirrhosis	2 (2.3)
Amyloidosis	1 (1.1)
Cholangiocellular carcinoma	1 (1.1)
Heterozygous Alpha-1-Antitrypsin-Deficiency	1 (1.1)
Budd Chiari	1 (1.1)
Immunosuppressive therapy at time of ERCP, absolute (%)	
Mycophenolate mofetil	163 (57.2)
Tacrolimus	237 (83.2)
Prednisolone	26 (9.1)
Cyclosporine	24 (8.4)
Sirolimus	14 (4.9)
Everolimus	77 (26.9)

**Table 2 jcm-13-01438-t002:** ERCP findings and intervention details; ITBL = ischemic type biliary lesions.

ERCP Findings, Absolute (%), *n* = 285	
Isolated anastomosis stenosis	175 (61.4)
ITBL	103 (36.1)
Choledocholithiasis (stone-free after intervention)	7 (2.5)
Interventions during ERCP, *n*	
Dilatation of bile duct	261
Papillotomy	82
Main bile duct stenting	42
Usage of Dormia basket	39
Cholangioscopy	38

**Table 3 jcm-13-01438-t003:** Infectiology results; MRSA = methicillin-resistant *Staphylococcus aureus*, VRE = vancomycin-resistant *Enterococcus*, 3MRGN = 3-multidrug-resistant gram negative.

Anti-infective Therapy, Absolute (%)	
Application of prophylactic anti-infective therapy 1 h before or during ERCP	188 (66)
Delayed application of prophylactic anti-infective therapy > 1 h after the ERCP	97 (34)
Piperacillin/tazobactam	210 (73.7)
Ceftriaxone	34 (12)
Meropenem	15 (5.3)
Ciprofloxacin	10 (3.5)
Other	16 (5.5)
Infectious complications, absolute (%)	
Infection after ERCP	46 (16.1)
Blood culture taken in case of infection	31 (67.4)
Blood culture positive	10 (32.3)
Blood culture results, absolute (%)	
*Enterococcus faecium*	3 (30)
*Staphylococcus epidermidis*	2 (20)
*Escherichia coli*	2 (20)
*Catabacter hongkongensis*	1 (10)
*Klebsiella oxytoca*	1 (10)
*Enterococcus casseliflavus*	1 (10)
MDR colonization at the time of ERCP, absolute (%)	110 (38.6)
VRE (anal swab)	20 (18.2)
3MRGN	18 (16.4)
- *Escherischia coli* (*n* = 11, 4× bile culture, 7× abdominal swab) - *Enterobacter cloacae* complex (*n* = 2, anal swab) - *Klebsiella pneumoniae* (*n* = 2, anal swab) - *Pseudomonas aeruginosa* (*n* = 2, abdominal swab) - *Citrobacter freundii* (*n* = 1, bile culture)	
MRSA (Nose/throat swab)	6 (5.5)
4 MRGN	4 (3.6)
- *Acinetobacter baumanii* (*n* = 1, anal swab) - *Enterobacter cloacae* (*n* = 1, anal swab) - *Klebsiella pneumoniae* (*n* = 1, abdominal swab) - *Serratia macescens* (*n* = 1, bile culture)	

**Table 4 jcm-13-01438-t004:** Univariable regression analysis results; OR = ddds ratio, ITBL = ischemic type biliary lesions, IQR = interquartile range, CI = confidence interval.

Characteristic	No Infectious Complication (*n* = 239)	Infectious Complication (*n* = 46)	OR (CI)	*p*-Value
Age at the time of ERCP, median (min./max.)	56 (21/77)	58 (22/71)	1.01 (0.98–1.04)	0.68
Sex, male, absolute (%)	166 (69.5)	44 (95.7)	9.62 (2.27–40.7)	0.002
Period from transplant to ERCP, months, median (min./max.)	18 (1/226)	14.5 (0/115)	0.99 (0.98–1.00)	0.14
Biliodigestive Anastomosis, absolute (%)	15 (6.3)	0 (0)	0.00 (0.00)	0.99
Immunosuppressive agent, absolute (%)				
Cellcept	138 (57.7)	25 (54.3)	0.87 (0.46–1.64)	0.67
Tacrolimus	202 (84.5)	35 (76.1)	0.58 (0.27–1.25)	0.16
Prednisolone	17 (7.1)	9 (19.6)	3.18 (1.32–7.66)	0.01
Cyclosporine	20 (8.4)	4 (8.7)	1.04 (0.34–3.21)	0.94
Sirolimus	12 (5.0)	2 (4.3)	0.86 (0.19–3.98)	0.85
Everolimus	64 (26.8)	13 (28.3)	1.08 (0.53–2.18)	0.84
Number of immunosuppressive agents, absolute (%)				
1 immunosuppressive agent	32 (13.4)	9 (19.6)	reference variable	
2 immunosuppressive agents	198 (82.8)	33 (71.7)	0.59 (0.26–1.35)	0.22
3 immunosuppressive agents	9 (3.8)	4 (8.7)	1.58 (0.39–6.35)	0.52
ERCP findings, absolute (%)				
Single anastomosis stenosis	159 (66.5)	16 (34.8)	reference variable	
ITBL	75 (31.4)	28 (60.9)	3.71 (1.89–7.27)	<0.001
Choledocholithiasis (stone-free after intervention)	5 (2.1)	2 (4.3)	3.98 (0.71–22.17)	0.12
ERCP intervention, absolute (%)				
Sphincterotomy	65 (27.2)	16 (34.8)	1.43 (0.73–2.79)	0.30
Dilatation	218 (91.2)	43 (93.5)	1.38 (0.39–4.83)	0.62
Cholangioscopy	26 (10.9)	12 (26.1)	2.89 (1.33–6.27)	0.007
Main bile duct stenting	35 (14.6)	7 (15.2)	1.05 (0.43–2.52)	0.92
Usage of Dormia basket	30 (12.6)	9 (19.6)	1.70 (0.74–3.86)	0.21
Periprocedural anti-infective therapy, absolute (%)	239 (100)	46 (100)	n.d.	n.d.
MDR colonization at the time of ERCP	90 (37.7)	20 (43.5)	1.28 (0.67–2.41)	0.46
Delayed application of prophylactic anti-infective therapy >1 h after ERCP	74 (31)	23 (50)	2.23 (1.18–4.23)	0.01
CRP after ERCP (mg/dL), median (IQR)	0.95 (0.5–2.7)	6.55 (3.9–10.0)	n.d.	<0.001
Leucocytes after ERCP (Tsd/μL), median (IQR)	4.95 (3.4–7.2)	6.27 (3.8–8.8)	n.d.	<0.001
Length of hospital stay, days, median (IQR)	4 (3–5)	5 (4–7)	n.d.	0.003

**Table 5 jcm-13-01438-t005:** Multivariable regression analysis results; OR = odds ratio, ITBL = ischemic type biliary lesions, CI = confidence interval.

Characteristic	OR (CI)	*p*-Value
Age at the time of ERCP	1.03 (0.99–1.1)	0.13
Sex	24.19 (4.36–134.21)	<0.001
Time from transplant to ERCP	0.99 (0.98–1.01)	0.59
LTx indication	0.99 (0.88–1.12)	0.86
Prednisolone	4.5 (1.55–13.04)	0.006
ITBL	4.51 (2.11–9.60)	<0.001
ERCP intervention		
Sphincterotomy	2.44 (1.05–5.67)	0.04
Dilatation	9.48 (1.24–72.33)	0.03
Cholangioscopy	3.22 (1.28–8.11)	0.02
Main bile duct stenting	1.13 (0.38–3.34)	0.83
Usage of Dormia basket	3.32 (0.94–111.75)	0.06
Delayed application of prophylactic anti-infective therapy > 1 h after the ERCP	2.93 (1.36–6.27)	0.006

**Table 6 jcm-13-01438-t006:** Multivariable Cox regression analysis results; OR = odds ratio, ITBL = ischemic type biliary lesions, CI = confidence interval.

Characteristic	OR (CI)	*p*-Value
Age	1.0 (0.97–1.07)	0.43
Sex	4.88 (0.86–27.69)	0.07
Prednisolone	0.97 (0.28–3.30)	0.96
ITBL	1.46 (0.55–3.84)	0.45
Cholangioscopy	3.15 (0.94–10.54)	0.06
Dilatation	0.45 (0.09–2.28)	0.36
Delayed application of prophylactic anti-infective therapy > 1 h after the ERCP	1.35 (0.47–3.9)	0.58
Infection after the first ERCP	5.23 (1.92–14.26)	0.001

## Data Availability

Data are unavailable due to patients privacy.

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
