# Peer review of "Risk Factors for Infectious Complications following Endoscopic Retrograde Cholangiopancreatography in Liver Transplant Patients: A Single-Center Study"

_jcm, 2024, doi:10.3390/jcm13051438_

Round 1

Reviewer 1 Report

Comments and Suggestions for Authors

This manuscript highlights risk factors for infections after ERCP in liver transplanted patients. It is a monocentric retrospective study, that elucidates infectious complications and demonstrates the importance of a periinterventional antiobiotic treatment.

There are a few issues to be improved or discussed:

1) In lines 25-27+120 of the abstract the authors describe a number of 175 detected anastomosis stenosis in 88 patients, what means, that they counted the same stenosis in one patient multiple times. I suggest to count each diagnosis regarding the patients and not the number of ERCP.

2) In the materials and methods paragraph, it is mentioned, that elevated CRP levels were considered as infection and those patients were excluded from the study. How did you distinguish between PSC patients with a low inflammation activity and an infection? Can you please discuss it in your manuscript?

3) The inclusion criteria aren't defined clearly. 

4) In line 77 one of the main inclusion criterion was the need for an elective ERCP. But the different indications of these elective ERCP were not listed. Please add them. It is hard to immagine, that none of the patients with ITBL or stenosis or choledocholithiasis was hospitalized due to an urgent ERCP indication.

5)Line 88: How is the CRP value increase defined? Is it just a singular value not significantly above the reference level or how much must the increase be?

6) In line 121 complications other than infections were excluded. For more transparency of the ERCP procedures it is quite crucial to mention also the other complications, that led to study exclusion.

7) In table1 in line 133 the indications for liver transplant aren't listed depending on the frequency. Please order the ranking of the indications.

8) Table 1: Rather to mention only the different immunosuppressive therapy, it would be better to list also how many double and triple immunosuppressive therapies occured and if a triple treatment had a higher infection rate than a double one.

9) In lines 142-143 the authors list 277 ERCP with intervention and 9 ERCP without an intervention, in sum it means 286 ERCP instead of 285. Please correct the wrong number.

10) Table 3 in line 168: If 260 out of 285 cases got the periinterventional antibiotic therapy and 97 were administered a delayed antibiotic therapy.How can it be that 97 cases instead of the missing 25 cases (285-260) were counted for delayed treatment? Please check the 97 cases and change in the table.

11) Lines 163-165: In which anatomical regions where the colonizations detected?

12) Table 3: How did you rank the different items in your table? Would you mind, to order them depending on the frequency?

13) Lines 175-176: Was the univariable logistic regression analysis of the delayed application of prophylactic anti-infective therapy done with the number of 97 or 25 (see comment no. 8)?

14) In table 2 the authors mention 262 cases of bile duct dilatation, while they show 261 cases of it in table 4 in the univariable regression analysis. Please recalculate with the right case number and correct the wrong case number.

15) In the discussion in lines 228-229 it is mentioned, that fever and/or an increased CRP values led to antibiotic therapy. How was the cause of the increased CRP value distinguished (infection and non infectious inflammation)? That definition means, that a CRP increase alone could have led to an antibiotic therapy, what is not recommended by an accurate antibiotic stewardship. Please discuss in your manuscript.

16) The manuscript doesn't mention, whether the ERCP were performed in a sterile area or not.

17) Lines 242-244: Is there an evidence, that unrestored contrast medium enhances the infection rate? The references ar very old. 

18) Too many references are older than 8 years and should be reviewed.

Comments on the Quality of English Language

Sometimes the authors use the term "gender" instead of sex. Please always use sex instead of gender.

Reviewer 2 Report

Comments and Suggestions for Authors

In the introduction section:

- Include a brief review of alternative management options

- what are some criteria used to decide on the use of prophylactic antibiotics

- distinguish/draw a concise line between general risk factors for ERCP-related complications and those that are elevated in liver transplant recipients

- detail the implications of incomplete biliary drainage in liver transplant patients

In the methods section:

- detail on the criteria for choosing between these tests in different scenarios

- justify the treatment of non-metric variables as categorical

- what are the likely implications of censoring patients with less than two years of follow-up

Results section is correctly presented

In the discussion section:

- I encourage the authors to propose a more standardized or comprehensive definition of infection post-ERCP based on the findings of this study

- compare the infection risk associated with cholangioscopy to that of other ERCP-related procedures

- what would be the reason that peri-interventional antibiotic prophylactic therapy did not influence infection rates

Conclusions are adequate.

References are adequate.

Title is adequate.

Round 2

Reviewer 1 Report

Comments and Suggestions for Authors

I want to thank the authors for the revised manuscript.